# Learnable Masks for Time Series Explainability using Time-Frequency Representations

Theresa Dahl Frehr[*1], Thea Brüsch[1], and Tommy Sonne Alstrøm[1]

[1]Department of Applied Mathematics and Computer Science, Technical University of Denmark
{tdafr, theb, tsal}@dtu.dk

## Abstract

The demand for explainable AI models continues to grow with the rise of AI-based solutions. Relatively few explainability methods address time series due to the complex nature of the data. We propose learning saliency maps over both time and frequency via the discrete wavelet transform and the short-time Fourier transform. Faithfulness scores show that our method is on par with current state-of-the-art methods.

## 1 Introduction

Several critical domains, such as climate [1], finance [2], and healthcare [3], heavily rely on time series data. With the increase of automated processes based on deep learning, the need for explainability of such models becomes increasingly more important. While a wide range of methods have been developed to explain predictions in image-based models, the same cannot be said for time series data.

Time series are difficult to interpret [4]. Despite many models being trained on the raw time signals, the important patterns may exist in a latent feature domain, such as the frequency domain [5].

Learnable masks are a popular choice for creating attribution maps in the time domain, making them a suitable option for explainable time series [6, 7]. The masks are learned through gradient descent using an objective function that masks out much of the input while not changing the model prediction. As these approaches assume that localized salient information is in the time domain, they fall short when relevance is found in the frequency domain.

To address this limitation, FreqRISE [8] was proposed as a model-agnostic framework that jointly learns salient features in both the time and frequency domains. The method estimates relevance using Monte Carlo sampling across multiple masks to identify key features. Although FreqRISE offers competitive performance to established baselines, such as Integrated Gradients [9] and Layer-wise Relevance Propagation [10], it suffers from inefficient sampling and can potentially introduce artefacts by zeroing out frequency components. To overcome

these issues, FLEXtime [11] was introduced to explain time series purely in the frequency domain by decomposing the signal into frequency bands using a filterbank.

In this work, we combine the strengths of FLEXtime and FreqRISE by introducing a mask-based approach to identify cross-domain saliency. By representing the signal jointly in the time and frequency domains, our method aims to learn a mask that captures salient features across time and frequency simultaneously.

## 2 Methods

We consider two signal representations: the Discrete Wavelet Transform (DWT) using the `Symlet2` mother wavelet, and the Short-Time Fourier Transform (STFT). The objective is to learn a mask, $M$, that highlights the most relevant components of the input signal $X$ for a given prediction task. The masked input is defined as the element-wise product between the time-frequency representation ($\tilde{X}$) of the input ($X$) and the mask: $X^M = \tilde{X} \circ M$. The prediction based on the masked input is then given by $\hat{y}^M = \boldsymbol{f}(X^M)$, where $\boldsymbol{f}(\cdot)$ represents the trained model. An optimal mask should preserve the model's predictive behaviour, i.e. $\hat{y}^M \approx \hat{y}$, where $\hat{y} = \boldsymbol{f}(X)$ is the original prediction. To measure the deviation caused by masking, we use the cross-entropy loss:

$$\mathcal{L}_D(\hat{y}, \hat{y}^M) = -\sum_{c=1}^{C} \hat{y}_c \log(\hat{y}_c^M). \qquad (1)$$

As $\hat{y}^M = \hat{y}$ if $M$ is all ones, we need to impose a sparsity constraint on the mask, which is done using the $\ell_1$-norm:

$$\mathcal{L}_R(M) = \max\left(\frac{\|M\|_1}{L} - r, 0\right), \qquad (2)$$

where $L$ is the number of elements in the mask and $r$ is a ratio parameter that controls the sparsity threshold. To ensure temporal consistency, we include a smoothness regularization term that penalizes abrupt changes in the mask values over time. For the STFT, where each coefficient corresponds to a uniform time interval, we use the following

---

*Corresponding Author.

**Table 1.** Hyperparameters.

|      | $\lambda_R$ | $\lambda_S$ | $r$  |
|------|------|------|------|
| DWT  | 0.5  | 10   | 0.1  |
| STFT | 2    | 5    | 0.05 |

smoothness loss:

$$\mathcal{L}_S = \frac{1}{L} \sum_{f \in \mathcal{F}} \sum_{t=1}^{T-1} (M_{f,t+1} - M_{f,t})^2, \qquad (3)$$

where $\mathcal{F}$ is the set of frequency bands. For the DWT, where time–frequency resolution varies across scales, a weighted version of this constraint is applied, assigning higher penalties to frequency bands with lower temporal resolution. The overall loss function thus becomes

$$\mathcal{L} = \mathcal{L}_D(\hat{y}, \hat{y}^M) + \lambda_R \mathcal{L}_R(M) + \lambda_S \mathcal{L}_S(M). \quad (4)$$

Evaluation of an explanation is a challenging task. As no ground truth exists, the quality of an explanation is usually determined by measuring different properties that are deemed desirable. In this work, we focus on faithfulness, which measures how aligned an explanation is with the prediction of a model. The idea is that by removing the 10% most important features as given by the explanation, the mean true class probability drastically drops. Likewise, by adding the 10% most important features, we expect it to be high.

## 3   Experimental Setup

Experiments are conducted on the AudioMNIST dataset [12], which consists of 30,000 audio recordings of spoken digits (0–9), each repeated 50 times by 60 different speakers (12 female/48 male). All recordings are downsampled to 8kHz and zero-padded to a fixed length of 8,000 samples, following the procedure from [12]. The 1D convolutional neural network proposed by [12] is used as the model $\boldsymbol{f}$. The network was trained and released in [8], and the same weights are used in this study to ensure comparability with previous work. The selected hyperparameters are listed in Table 1. For the DWT representation, the mask is initialized with 10% non-zero values per frequency band, drawn uniformly from $[0, 0.1]$. For the STFT representation, 10% of elements are randomly distributed across time and frequency with values from $[0, 1]$.

Experiments are reported as an average of 300 samples run over 50 different seeds, which can be seen in Table 2. For comparison, FreqRISE gets an insertion faithfullness score of 0.86 and FLEXtime of 0.91 [11]. In Figure 1, an example explanation is shown for the time series corresponding to the spoken digit "4" (top row), using both the DWT (second

**Table 2.** Mean faithfulness scores across 50 experiments. I = insertion, D = deletion, RI = random insertion, RD = random deletion.

|      | I($\uparrow$) | D($\downarrow$) | RI($\uparrow$) | RD($\downarrow$) |
|------|------|------|------|------|
| DWT  | 0.91 | 0.63 | 0.25 | 0.95 |
| STFT | 0.96 | 0.77 | 0.41 | 0.96 |

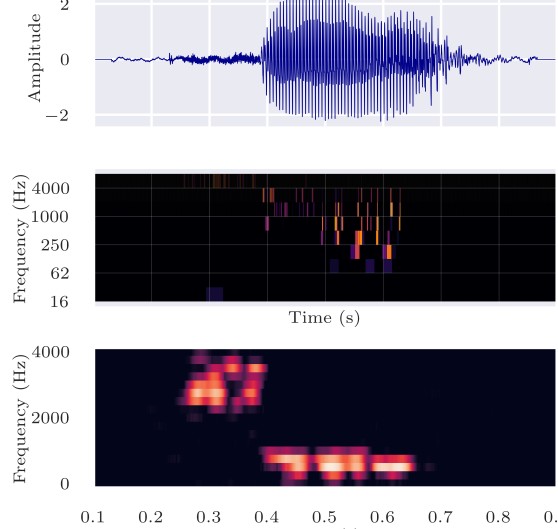

**Figure 1.** **Top**: Time series of the digit "4". **Middle**: Learned attribution map DWT. **Bottom**: Learned attribution map using STFT.

row) and STFT (third row) representations. Both methods highlight higher-frequency components in the interval $[0.2, 0.4]$ s and lower-frequency components in the interval $[0.4, 0.65]$ s. The mask obtained with the DWT appears less smooth compared to the STFT.

## 4   Discussion and Conclusion

We proposed a framework for learning saliency maps in the time–frequency domain using different signal representations. Both the STFT-based and the DWT-based approaches achieved high insertion faithfulness scores, with STFT performing slightly better and even surpassing established baselines such as FreqRISE and FLEXtime. The attribution maps generated from the DWT representation appeared less smooth and displayed more jitter. This may be due to the non-uniform resolution of the DWT. This suggests that the fixed time–frequency resolution of the STFT may be more suitable for audio data.

Future work will focus on optimizing hyperparameters, extending the evaluation to additional datasets and metrics, and conducting a more systematic comparison with other explainability methods.

# Acknowledgments

This work was supported by the William Demant Fonden under grant number 24-5326.

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
