# OpenReview forum: "Learnable Masks for Time Series Explainability using Time-Frequency Representations"
_NLDL.org/2026/Abstracts_Track — NLDL 2026 Abstracts_

### Official Review · Reviewer_ixbe · 2025-10-24

**Soundness:** 3
**Correctness:** 3
**Rating:** 5
**Confidence:** 4

**Summary:**

The paper proposes an explainability method for time series models that learns saliency maps in both time and frequency domains using wavelet and short-time Fourier transforms, improving interpretability and achieving state-of-the-art faithfulness performance.

**Strengths:**

1. Relevant problem and novel contribution to the research community
2. Model-agnostic framework
3. Convincing preliminary results on AudioMNIST dataset
4. Clear future work directions

**Weaknesses:**

1. Experimental results section lacks comparison with the related work.
2. Evaluation is needed for additional datasets

---

### Official Review · Reviewer_E1E2 · 2025-10-30

**Soundness:** 3
**Correctness:** 3
**Rating:** 4
**Confidence:** 3

**Summary:**

The authors propose an explainability method for time-series data using learnable masks. Their method takes both frequency and time domains into account and proposes to learn a mask that captures saliency features in both domains simultaneously. They experiment with two forms of input data: STFT and DWT, and report faithfulness scores for I = insertion, D = deletion, RI = random insertion, and RD= random deletion.

**Strengths:**

The motivation for combining both time and frequency seems clear. The paper proposes a clear loss function comprising three parts. Figure 1 provides an interesting visualization of the method, making the contribution even more understandable.

**Weaknesses:**

The authors mention that “Faithfulness scores show that our method is on par with current state-of-the-art methods.”. However, they do not compare their method to FreqRISE and FLEXtime, which they mentioned as a baseline. I would suggest comparing baseline methods on their poster, even if the results are not good.
Despite the weakness, I think that the idea and motivation behind this abstract are clear and sound, and will spark fruitful discussions at the poster session that will help the authors complete their work into a full paper.

---

### Official Review · Reviewer_PZC5 · 2025-11-03

**Soundness:** 4
**Correctness:** 4
**Rating:** 5
**Confidence:** 4

**Summary:**

The paper concerns explainability on time series data, and proposes a method combining representations in the time and frequency domain, building on previous work where only one or the other domain has been used. They consider both a Discrete Wavelet Transform and Short-Time Fourier Transform. The goal is then, for a pair of input and prediction, to learn a mask such that the prediction changes minimally, which captures that the masked features are less relevant. The method is evaluated by faithfullness score, and an example explanation is provided.

**Strengths:**

The writing is very clear and easy to follow. The method is a clear extension of previous work that shows promising results, and the problem is relevant for the community. It is pointed out that evaluating explanations is not straight forward and that there are several other ways to do it, which avoids overselling the results. Most weeknesses are already covered as suggested future work.

**Weaknesses:**

The choice of hyperparameters are not motivated, nor is there any discussion of wether these are expected to change between applications or how much they affect the results. As the problem is motivated by high stakes decisions in finance and climate, evaluation on a related data set would have strengthened the paper. Including the previous methods directly in Table 2 could improve readability.

---

### Decision · Program_Chairs · 2025-11-05

**Decision:**

Accept

**Comment:**

The abstract is of interest to the community and should be presented at the conference.